# Assessment of Anti-Bacterial Effectiveness of Hand Sanitizers Commonly Used in South Africa

**DOI:** 10.3390/ijerph19159245

**Published:** 2022-07-28

**Authors:** Lufuno Muleba, Renay Van Wyk, Jennifer Pienaar, Edith Ratshikhopha, Tanusha Singh

**Affiliations:** 1National Institute for Occupational Health, National Health Laboratory Service, Johannesburg 2000, South Africa; lufunom@nioh.ac.za (L.M.); edithr@nioh.ac.za (E.R.); 2Department of Biomedical Sciences, Faculty of Health Sciences, University of Johannesburg, Johannesburg 2028, South Africa; 3Department of Environmental Health, Faculty of Health Sciences, University of Johannesburg, Johannesburg 2028, South Africa; renayvw@uj.ac.za; 4Biomedical Sciences, Faculty of Health and Life Sciences, De Montfort University, Leicester LE1 9BH, UK; jennifer.pienaar@dmu.ac.uk; 5Department of Clinical Microbiology and Infectious Diseases, University of Witwatersrand, Johannesburg 2050, South Africa

**Keywords:** hand disinfectants, microorganisms, hospital-acquired infection, infection prevention and control, precautionary measures, hand hygiene

## Abstract

Hand sanitizers are used as an alternative to hand washing to reduce the number of viable microorganisms when soap and water are not readily available. This study aimed to investigate the anti-bacterial effectiveness of commercially available hand sanitizers and those commonly used in healthcare and community settings. A mapping exercise was done to select and procure different hand sanitizers (*n* = 18) from retailers. Five microorganisms implicated in hospital-acquired infections were selected and tested against each hand sanitizer: *Escherichia coli*, *Enterococcus faecalis, Klebsiella pneumoniae, Pseudomonas aeruginosa*, and *Staphylococcus aureus*. Twenty-one volunteers were recruited to do a handprint before and after applying the hand sanitizer. Only four out of eighteen hand sanitizers (22%) were effective against all tested bacterial species, and an equal number (22%) were completely ineffective. Seven hand sanitizers with a label claim of 99.99% were only effective against *E. coli*. Only five hand sanitizers (27%) effectively reduced bacteria on participants’ hands. This study showed that only a fifth of hand sanitizers were effective against selected microorganisms. The findings raise a concern about the effectiveness of hand sanitizers and their role in infection, prevention, and control if not well regulated.

## 1. Introduction

The emergence of novel microbial pathogens has presented major public health challenges globally. Some of these pathogens are resistant to a range of common antibiotics; refs. [1,2] highlighting prevention of transmission as the best method to reduce exposure and contain major outbreaks in workplaces and community settings. Furthermore, the explosion of products containing antibacterial agents defies the critical message that washing with soap and water is sufficient to provide hygiene to healthy individuals. A growing number of microorganisms are emerging with resistance to these chemicals leading to enhanced morbidity and mortality; hence the term “super bugs” was coined. The inappropriate use of antimicrobial drugs, inadequate sanitary conditions, inappropriate food handling, ineffective disinfection, and poor infection prevention and control practices are determinants of the microbial resistance phenomenon [3]. Although frequent hand washing with soap and water is a cost-effective hand hygiene tool that can be implemented to prevent the spread of pathogenic microorganisms [4], hand sanitizing with alcohol-based hand sanitizers (ABHS) and non-alcohol-based hand sanitizers (NABHS) has become a preferred alternative, particularly where the practice is impractical or where water is unavailable. Hand sanitizing is a key precautionary measure and has become “a norm” due to the coronavirus disease of the 2019 (COVID-19) pandemic. Various hand sanitizers have been used and continue to be used in healthcare, non-healthcare, and community settings to prevent the spread of bacterial infection due to hand contamination [5,6].

The antimicrobial agents in hand sanitizers differ, with NABHS commonly comprising chlorohexidine, benzalkonium chloride, iodophors, or quaternary ammonium compounds, and triclosan, all of which are used in various formulations and combinations [7,8]. On the contrary, alcohol-based hand sanitizers have formulations or combinations that may include ethanol, n-propanol, and isopropanol. Some inactive ingredients such as sterile water, polyethylene, polyacrylic acid, glycerin, hydrogen peroxide propylene glycol, and plant extract are added to hand sanitizer formulations. These ingredients help to reduce the drying and damaging effect of alcohol on the skin and increase the biocidal activity of sanitizers by prolonging the time needed for evaporation of alcohol [9,10]. These ABHS are most effective against an array of microorganisms at an alcohol concentration of 60–90% [11].

The active mechanism of hand sanitizers against bacteria includes protein denaturation, inhibition of mRNA, and protein synthesis [12,13]. Different ABHS have demonstrated antimicrobial effects against various Gram-positive (e.g., *Enterococcus faecium, S. aureus*, vancomycin-resistant Enterococcus) and Gram-negative (e.g., *E. coli*, *P. aeruginosa*) bacteria [11,14,15]. In addition, some studies have also shown that propanol-based sanitizers are more effective than ethanol-based sanitizers [8,16]. In South Africa, hand sanitizers are not well regulated, and the product’s or manufacturer’s claims of microbial effectiveness are not verified. Hence, ineffective products enter the market leading to a false sense of security, as highlighted by a recent study that evaluated the alcohol content of hand sanitizers [17]. Due to a paucity of information on microbial effectiveness, this study aimed to investigate the anti-bacterial effectiveness of hand sanitizers commonly used in healthcare and community settings in Johannesburg, South Africa.

## 2. Materials and Methods

### 2.1. Study Design

The study included an experimental design and cross-sectional design conducted at the National Institute of Occupational Health, Immunology and Microbiology Laboratory, South Africa. The cross-sectional design was used to determine the effectiveness of hand sanitizers in reducing bacterial concentration on the hands of volunteering laboratory staff (laboratory assistants, intern medical scientists, medical scientists, and receiving officers) from a large laboratory service setting. All participants, which included 8 males and 13 females older than 18 years consented to participate.

### 2.2. Sterility Testing of Hand Sanitizers

Popular brands of hand sanitizers randomly purchased from various traders were first tested for contamination before assessing the anti-bacterial effectiveness. Three of the eighteen hand sanitizers were from the same manufacturing company. Information regarding the sanitizers’ ingredients, application and kill rate, approval body, and application was recorded. See Appendix A.

### 2.3. Bacterial Strains

Five bacterial strains, namely *Escherichia coli* American Type Culture Collection (ATCC 29220 (*E. coli*), *Enterococcus faecalis* ATCC 29212 (*E. faecalis*), *Klebsiella pneumoniae* NCTC 9633 (*K. pneumoniae*), *Pseudomonas aeruginosa* ATCC 27853 (*P. aeruginosa*), and *Staphylococcus aureus* ATCC 25923 (*S. aureus*), that are known to cause nosocomial infections, were selected to investigate the effectiveness of hand sanitizers against these agents. The bacterial strains were obtained from the national stock culture collection, reconstituted, and maintained according to the supplier’s instructions before being inoculated onto nutrient agar (NA) (Diagnostic Media Product, South Africa) and incubated at 37 °C overnight. A bacterial suspension containing 2 mL of sterile normal saline (NaCl 8.5 g, distilled water 1 L) was aseptically made for the five strains and was adjusted to an optical density of 0.5 McFarland standard [18]. 

### 2.4. Disk Diffusion Method

The microorganisms were evenly spread on the surface of 90 mm Mueller–Hinton Agar plates (Diagnostic Media Product, South Africa) using a cotton swab and allowed to dry for approximately 5 min inside the Biosafety cabinet class (BSC) II (Lab and Air, South Africa). A sterile 4 mm cork borer (Lasec, South Africa) was used to bore four holes approximately 25 mm apart from each other in the agar. One hundred microliters of the selected hand sanitizer were pipetted into three holes, while sterile water was pipetted into one hole to serve as a negative control. For the positive controls, a separate plate was used where 100 microliters of laboratory-prepared absolute 70% ethanol were pipetted into a hole, and the streptomycin antibiotic paper discs were placed on the agar plate with boring a hole. This was repeated for all the test organisms and hand sanitizers. The agar plates were then incubated for 18–24 h (hr) at 37 °C. Zones of inhibition showed the measure of resistance or susceptibility (sensitivity) of the test organism to the anti-bacterial agent and were measured with the aid of a digital caliper (mm) (Mantech Electronics, Johannesburg, South Africa). For the efficacy assessment, bacterial strains that showed zones of inhibition greater than or equal to the standard zones (70% ethanol and 25 micrograms (mcg) streptomycin disc) were considered to be sensitive to the hand sanitizer; however, if the zone was smaller, the tested microorganism was considered to be resistant to the hand sanitizer [11,19].

### 2.5. Determination of the Minimum Inhibitory Concentration (MIC) and Minimum Bactericidal Concentration

To determine the MIC, one milliliter of standardized culture (0.5 McFarland) of each bacterial strain was mixed with one milliliter of each hand sanitizer and incubated for 18–24 h at 37 °C and observed for turbidity. A loopful of inoculum from the MIC tubes (vortexed for 2 min, 30 s) was streaked on nutrient agar plates using a sterile loop and incubated (Labotec, Midrand, South Africa) at 37 °C for 18–24 h and observed for growth; no growth. The negative and positive controls were also included.

### 2.6. Assessment of the Effectiveness of Hand Sanitizers by Handprint Method

Participants (*n* = 21) were informed not to wash or sanitize their hands 60 min before their morning and afternoon testing. There was a minimum 3.5 hr gap between the morning and afternoon sessions. The participant applied the hand sanitizer according to the manufacturer’s instructions displayed on the container. Handprints were done on 5% blood agar plates (Diagnostic Media Product, Johannesburg, South Africa) before and after applying a hand sanitizer and incubated at 37 °C for 72 h. For products without instructions, participants applied the sanitizer according to their knowledge and hand-printed. Each participant acted as their control by doing a handprint pre- and post-application of the hand sanitizer (Table 1). The bacterial growth was graded based on colonies that grew on agar. Agar plates with zero colonies were rated as no growth observed (0+), 1–10 colonies were rated as low growth (1+), 11–20 colonies were rated as medium growth (2+), and plates with more than 20 colonies were rated as high growth (3+).

### 2.7. Statistical Analysis

Data was captured and cleaned in Microsoft Excel 2016 (Microsoft, Washington, DC, USA) and exported to Stata version 15.2 (StataCorp, College Station, TX, USA). Statistical analysis was carried out using Stata version 15.2 software. Numerical variables were summarized using means and standard deviations (SD). The non-parametric data using the Dunns test assessed the performance difference of the hand sanitizers in reducing the bacterial counts on participants’ hands. This was determined by collecting the categorical data of the handprints of participants and analyzing using the test Pearson’s chi-squared test (Chi^2^) statistic [20]. A 5% significance level representing a *p* < 0.05 to reject the null hypothesis of no association was used. We followed STROBE guideline recommendations.

## 3. Results

Sterility tests confirmed the absence of microbial contamination of the hand sanitizers before the experiments. The differences in the means of various groups were compared using ANOVA Microsoft Excel 2016. There were no significant differences in the ZOI for different sanitizers tested against the bacterial strains used for testing (*p* = 0.2113). However, there were variations within each bacterium tested with the highest for *E*. *coli* and S. *aureus*; the SDs were 4.16 and 4.17, respectively. This was followed by *E. faecalis* (SD = 3.83). The smallest variations were observed for *K. pneumoniae* and for *P. aeruginosa* with SDs of 2.73 and 2.56, respectively (Figure 1). Figure 1 also shows the ZOI for individual bacteria for each sanitizer tested. There were significant differences in the average ZOI for the five bacteria tested by different sanitizers used (*p* = 0.0000). The highest average ZOI was for hand sanitizer G (13.43 mm) and hand sanitizer A (12.09 mm) and the lowest were for hand sanitizer D (1.57 mm) and hand sanitizer P (1.17 mm). Streptomycin and 70% ethanol were used as laboratory positive controls or standards. If any of the tested sanitizer’s ZOI is greater than the controls, it is regarded as effective, and a ZOI less than these controls is regarded as less effective. Sanitizers D, K, P, and Q were not effective in reducing the growth of all bacterial strains. Hand sanitizers A, G, and R were effective against all bacterial strains as their zone of inhibition was greater than the two positive controls, however, *P*. *aeruginosa* and *K*. *pneumonia* showed resistance to hand sanitizers I and M, respectively. When comparing the results, the 70% ethanol control and hand sanitizers A, G, and R were effective in inhibiting the growth of all bacterial strains (Figure 1). Hand sanitizer P showed no ZOI when tested with *E. faecalis* and *S. aureus.* Hand sanitizer D showed no zone of inhibition for *E. coli*. The concentration of alcohol was not indicated on the ingredients of hand sanitizers (D, E, F, J, K, L, N, O, P, Q, R), whereas hand sanitizer E had no alcohol. 

### 3.1. Minimum Inhibitory Concentration (MIC) and Minimum Bactericidal Concentration (MBC)

The MIC test showed that only five of eighteen (28%) hand sanitizers passed, i.e., showed no visible growth with all bacterial strains. Thirteen (72%) of the hand sanitizers showed visible growth with all bacterial strains, except with *K. pneumoniae* where only four (22%) showed no growth.

The minimum bactericidal concentration (MBC) is the lowest concentration of an anti-bacterial agent required to kill a particular bacterium over 18 or 24 h and is reported as growth or no growth. The MBC was observed with only one hand sanitizer E; it had too numerous to count colonies. For quality purposes, all controls passed; the negative control (sterile water) had no growth, and the positive controls (standardized culture of all bacterial strains) had growth on the agar plates.

### 3.2. Assessment of the Effectiveness of Hand Sanitizers by Handprint Method

A total of 21 laboratory workers voluntarily participated in the study, and 20 completed testing of hand sanitizers A to R. One participant reacted to hand sanitizers (M) on day 1 of testing and thus withdrew from the study. All hand sanitizers were well endured, and there were no further complaints reported of skin irritation or other symptoms. The participants ranged from 23 to 60 years, with a mean age of 37.75 years (SD = 11.173). There were no significant differences in the bacterial growth (no growth, 1+ growth, 2+ growth, 3+ growth) from handprints between different age groups (21–30, 31–40, 41–50, 51–60), *p* > 0.05. There were also no significant differences in the bacterial growth (no growth, 1+ growth, 2+ growth, 3+ growth) from handprints by sex (*p* > 0.05).

The handprints of all participants had bacterial growth before hand sanitizer application. A high percentage of participants (>50%) had growth after using hand sanitizers. The difference in the number of participants who had bacterial growth in handprints after using different sanitizers was significant (*p* < 0.001). Hand sanitizer G had 0% growth after the participants applied the hand sanitizer whereas hand sanitizers D, K, L, O, P, and Q had 100% bacterial growth post handprints. The difference was observed mainly between hand sanitizers G, H, and N (which had less than 50% of participants with growth) and all other hand sanitizers (*p* < 0.005). The number of participants with bacterial growth after using hand sanitizer G was significantly different from after using most other hand sanitizers (B, C, D, F, J, K, L, M, O, P, Q, and R). There were significant differences in the level of bacterial growth per rated category (0+, 1+, 2+, and 3+) after using different hand sanitizers (*p* < 0.005).

## 4. Discussion

One of the most effective ways to prevent the spread of pathogens, including bacteria, in workplace and community settings is hand hygiene using hand sanitizers. The use of hand sanitizers has gained popularity in the world in the past years as a result of the COVID-19 pandemic, which mainly uses supportive and preventative ways to reduce transmission [13,21]. This has led to the development, production, and importation of several hand sanitizers in South Africa by various traders. With the increased use of hand sanitizers, there is a need to assess the effectiveness of the products available in the market. This study investigated the anti-bacterial effectiveness of hand sanitizers commonly used in South Africa.

The hand sanitizers in this study were used in their original state by the manufacturer/producers (the alcohol concentrations were not modified). Sterility tests were carried out on all the hand sanitizers (A–R) and showed no microbial contamination. These results are reassuring and did not negatively influence the outcome of the experiments. Previous studies have not reported the sterility of the hand sanitizers before testing and hence is unique to this study. The sanitizers exhibited variable inhibitory activity against the bacterial strain, with zones of inhibition ranging from 1.5 mm to 14.94 mm at the manufacturer’s concentrations. In general, the current study results showed lower effectiveness of tested products with lower zones of inhibition than previously published [22,23,24]. The reduced effectiveness may be indicative of consumer demand and compromised manufacturing practices.

A study [23] tested four waterless gel forms of ABHS against *E. coli*, *P. aeruginosa*, *K. pneumoniae*, and *S. aureus*. In that study, handprints were done by five participants using two hand sanitizers. The highest zone of inhibition was observed with *S. aureus* (12 mm), which was similar to our study, where the highest zone of inhibition recorded was with *S. aureus* (14.82 mm). Conversely, *P. aeruginosa* was the most resistant organism when comparing the zone of inhibition with 70% ethanol. The most resistant bacterial strain in the current study was *K. pneumonia* and *P. aeruginosa*. These results suggest that certain hand sanitizers or decontaminants may not be effective against all bacteria. Hence, assessing the effectiveness of hand sanitizers against common pathogens before marketing is vitally important.

An earlier study [24] tested the effectiveness of three hand sanitizers against *E. coli*, *K. pneumoniae*, *P. aeruginosa*, and *S. aureus* and found a higher zone of inhibition for *K. pneumonia* and *S. aureus* when compared with the 70% ethanol positive control. In this study, only one hand sanitizer (D) had a lower zone of inhibition when tested against *K. pneumoniae*, and one hand sanitizer (P) had low inhibition for *S. aureus*. The lower zone of inhibition was observed between *K. pneumoniae* and *S. aureus* when compared with streptomycin 25 mcg. The lower zones of inhibition in this study mean that the bacterial strain can grow despite the hand sanitizers, highlighting poor effectiveness and thus compromising hand hygiene as a precautionary measure. Despite different brands having different formulations, the hand sanitizer must effectively reduce microbial contamination on the hands. Hand sanitizer D had no zone of inhibition against *E. coli*, and hand sanitizer P had no zone of inhibition against *E. faecalis* and *S. aureus*. The inability of these hand sanitizers to inhibit the growth of these organisms has implications in hand hygiene since these organisms are notorious for their ability to cause a wide variety of diseases and have a high level of antibiotic resistance [25]. Hand sanitizers A, G, I, and R were the most effective against all tested bacterial strains. Several hand sanitizers (*n* = 8) with a label claim of 99.99% effectiveness were effective against some but not all five bacterial strains tested. Differences in the level of effectiveness of hand sanitizers on the market have been reported previously in other studies [11,24]. A study in Pakistan [18] tested three hand sanitizers against 31 microorganisms isolated from currency notes and coins and reported that one of the products was only effective against 6.5% (*n* = 31) of the microorganisms tested. Another study in Kenya [11] tested 14 hand sanitizers against non-pathogenic strains of *E. coli*, *S. aureus*, and *P. aeruginosa*. The participants (*n* = 10) artificially contaminated their hands and only 25% of hand sanitizers were effective against a third (33%) of the microorganisms tested. The same study also noted that the difference in efficacy of the various hand sanitizers could arise from the actual composition of alcohol present in the product. Although this study did not focus on the alcohol content, two hand sanitizers (B and H) that claimed to be 99.9% effective had ethanol and ethyl as active ingredients. No test was done in this study to confirm if the actual alcohol content is the same as indicated on the label. Transient microorganisms can survive and sporadically multiply on skin surfaces including the hands. Most of these organisms are associated with hospital-acquired infections such as *Staphylococcus epidermidis, Clostridium difficile*, methicillin-resistant *Staphylococcus aureus* (MRSA), vancomycin-resistant *enterococci* (VRE), and multidrug-resistant (MDR) gram-negative rods including *Acitonobacter baumannii* [26]. Alcohol-based sanitizers work by removing the oil layer on the surface of the skin, thereby inactivating the microorganisms present. Transient microbial regrowth is slowed after sanitizing effectively, keeping “residual” organisms within the deeper layers of skin from coming to the surface [27].

However, local researchers found that 41% (37/94) of the alcohol-containing sanitizers contained less than 60% *v*/*v* alcohol. Ethyl acetate, isobutanol, and other non-recommended alcohols (methanol, 1-propanol, and 3-methyl-butanol) were detected [17]. Alcohol components are the major active ingredients in most ABHSs, where they disrupt the cell wall and or cell membranes by denaturing proteins and dissolving lipids [22]. In the current study, hand sanitizer G showed the highest zone of inhibition across all species. This means that hand sanitizer G can effectively inhibit the growth of bacteria tested. Another study [24] showed that hand sanitizers with higher alcohol concentrations (up to 90%) are more effective than products with lower alcohol concentrations (below 60%), explaining why hand sanitizer G was effective against the test bacteria in this study. Additionally, hand sanitizer G recommends using the largest volume of 2.5 mL–5 mL.

The findings from the minimum inhibitory concentration tests showed that 5 (28%) hand sanitizers showed no visible growth with all bacterial species tested, whereas 13 (72%) of the hand sanitizers showed visible growth with all species. Hand sanitizer E had poor bactericidal activity against all test organisms, as all agar plates showed visible colonies. These results suggest that hand sanitizer E does not contain the required levels of antimicrobial agents sufficient to kill bacteria. The World Health Organization (WHO) also recommends hand sanitizers with a concentration of at least 60% ethanol and 70% isopropanol for the greatest germicidal efficacy [28]. Hand sanitizer E does not meet the recommendation made by WHO, as it contains less than 70% alcohol [28]. The hand sanitizers used in the study with less than 70% alcohol or no recorded alcohol percentage showed better bacterial reduction than hand sanitizer E. Furthermore, its ineffectiveness in killing bacteria illustrates the importance of having the correct antimicrobial formulation at a concentration to assure product confidence. This prevents a false sense of security for end-users such as workers and the general public.

Human skin provides nutrients that have growing conditions conducive for most pathogens and opportunistic bacterial pathogens. The current study shows that certain tested hand sanitizers reduced the bacterial load on the hands of the participants to varying degrees. Hand sanitizers without indicated alcohol concentrations (D, K, L, O, P, Q, and R) were poorly effective in reducing growth post-application with a lower overall bacterial log reduction. The study has similar findings Ochwoto et al. (2017), where 28% of the hand sanitizers containing more than 60% isopropyl alcohol were more effective than 50% of hand sanitizers with less than 60% isopropyl alcohol. Another study [23] tested four different hand sanitizers using the handprint method and found that the bactericidal activity of all the hand sanitizers failed to achieve 99.9% killing of bacteria as was claimed on their labels. However, all the hand sanitizers [23] had no alcohol as the active ingredient. A study [5] reported that alcohol-based hand rubs containing n-propanol or isopropanol showed significantly greater skin irritation than ethanol-based ones. Therefore, individuals may tend not to use it and thus limit the effectiveness of the hand hygiene programs as they react negatively to hand sanitizers [5]. Only one participant complained of skin irritation in the current study and withdrew from the study on the first day of participation. The skin reaction could be related to alcohol denat and ethyl alcohol, active ingredients. Age and sex may be associated with knowledge and attitudes regarding the use of hand sanitizers. However, the current study did not find any significant differences among the participants. This may be due to the small sample size or that the participants in the study had a similar level of understanding and attitude regarding the use of hand sanitizers.

In contrast, in the current study, hand sanitizer G showed high bactericidal activity; no growth was found in all handprint agar plates of the participants’ post-application of the hand sanitizer. There was growth in all agar plates before applying hand sanitizer G, and no growth after applying the hand sanitizer. A study [9] tested 7 different hand sanitizers on 21 participants. Amongst these sanitizers, two hand sanitizers had a mean bacterial load reduction of 100% on the handprints. These two hand sanitizers had 0.5% chlorohexidine as one of their active ingredients [9]. These results concur the findings in the current study, that complete bactericidal activity by hand sanitizer G with similar chemical ingredients (0.5% chlorhexidine in 70% alcohol) achieved similar bacterial load reduction. Hand sanitizer A also contains chlorhexidine; however, the percentage and application volume are not specified. In addition to the concentration of the active ingredient alcohol, the type of alcohol in hand sanitizers has implications for the individuals using them.

The South African Bureau of Standards (SABS) has two standards, South African National Standard (SANS) 490 and SANS 1853, that describe the testing and certification of ABHS [29,30]. The SANS 490 indicates that 3 mL of the hand sanitizer must be applied to the cupped hand [29]. The volume of the hand sanitizer to be applied is crucial as less volume applied might not cover the entire surface area of those with big hands. This might affect the delivery of the microbial effect of the hand sanitizer, exposing the individual to pathogens. In this study, only two (G and M) of the 18 hand sanitizers met the SANS 490 volume criteria for application. It is important to ensure that all products meet these standards before registering with the South African Health Products Regulatory Authority (SAHPRA). This will reduce the number of ineffective products on the market. During the study, hand sanitizer G was registered with SAHPRA. Hand sanitizer H was certified by the South African Bureau of Standards (SABS), and approved using the SANS 490.

On the other hand, 15 (83%) of hand sanitizers were neither registered with the SAHPRA nor SABS, thus raising questions about the regulation of these products. The hand sanitizers from the current study were tested according to the manufacturer’s instructions. Based on the findings, the manufacturer should specify the volume to be used on the container, e.g., apply one or two pumps or squeeze the container twice or three times.

The limitation of the study was the small sample size. Further research can account for this if time and funding are available.

## 5. Conclusions

The results of this study concur with previous studies but are alarming as product effectiveness has not improved over time. The performance of several brands that indicated 99.9% effectiveness did not perform as well as the claims from the manufacturers. This study showed that only a fifth of the hand sanitizers tested were effective against selected microorganisms and warrants further investigation into labeling claims. The findings substantiate the need to regulate this industry as it plays a critical role in infection, prevention, and control in workplace and community settings.

## Figures and Tables

**Figure 1 ijerph-19-09245-f001:**
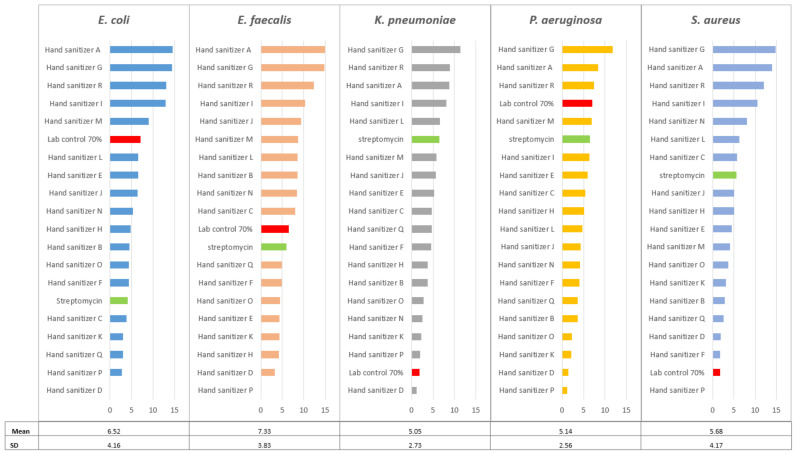
Zones of inhibition (mm) of the bacterial strains against hand sanitizers (A–R). The positive control is reflected by the green highlight, and the zone of inhibition lower than the positive control is shown by the red highlight.

**Table 1 ijerph-19-09245-t001:** Percentage of microbial growth pre- and post-hand sanitizer application by laboratory workers.

Sample Identification	Number of Participants that had Percentage (%) Growth				
	0^+^ Growth	1^+^ Growth	2^+^ Growth	3^+^ Growth	Any Growth *
Hand sanitizer A	11 (55%)	9 (45%)	0 (0%)	0 (0%)	9 (45%)
Hand sanitizer B	6 (30%)	14 (70%)	0 (0%)	0 (0%)	14 (70%)
Hand sanitizer C	4 (20%)	11 (55%)	3 (15%)	2 (10%)	16 (80%)
Hand sanitizer D	0 (0%)	0 (0%)	11 (55%)	9 (45%)	20 (100%)
Hand sanitizer E	2 (10%)	15 (75%)	3 (15%)	0 (0%)	18 (90%)
Hand sanitizer F	6 (30%)	11 (55%)	3 (15%)	0 (0%)	14 (70%)
Hand sanitizer G	20 (100%)	0 (0%)	0 (0%)	0 (0%)	0 (0%)
Hand sanitizer H	13 (65%)	5 (25%)	2 (10%)	0 (0%)	7 (35%)
Hand sanitizer I	10 (50%)	7 (35%)	3 (15%)	0 (0%)	10 (50%)
Hand sanitizer J	7 (35%)	10 (50%)	3 (15%)	0 (0%)	13 (65%)
Hand sanitizer K	0 (0%)	12 (60%)	8 (40%)	0 (0%)	20 (100%)
Hand sanitizer L	0 (0%)	10 (50%)	10 (50%)	0 (0%)	20 (100%)
Hand sanitizer M	6 (30%)	11 (55%)	3 (15%)	0 (0%)	14 (70%)
Hand sanitizer N	12 (60%)	5 (25%)	3 (15%)	0 (0%)	8 (40%)
Hand sanitizer O	0 (0%)	9 (45%)	10 (50%)	1 (5%)	20 (100%)
Hand sanitizer P	0 (0%)	2 (10%)	13 (65%)	5 (25%)	20 100%)
Hand sanitizer Q	0 (0%)	7 (35%)	13 (65%)	0 (0%)	20 (100%)
Hand sanitizer R	3 (15%)	11 (55 %)	6 (30%)	0 (0%)	17 (85%)

* Any growth = a combination of 1^+^, 2^+^, and 3^+^ growth.

## Data Availability

Data is available upon reasonable request and within the prescripts of the Protection of Personal Information Act (POPIAct).

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
