# Peer review of "Assessment of Anti-Bacterial Effectiveness of Hand Sanitizers Commonly Used in South Africa"

_ijerph, 2022, doi:10.3390/ijerph19159245_

Round 1

Reviewer 1 Report

Authors used an intervention study to assess the anti-bacterial effectiveness of hand sanitisers used in South Africa. Because use of hand sanitisers is believed as an alternative to hand washing, it is meaningful to assess the effectiveness in real world settings.  However, this article has not fully answered some of the questions due to insufficient description.

First, authors conducted an intervention study of 21 participants (P3L124), but the basic characteristics (e.g., gender and age) were not described. Because these factors may affect the effectiveness, it is important to disclose the information. Authors should add the statistics of basic characteristics of participants in the method section or results section.

Second, some of the paragraphs explaining the method are described in the result section (e.g., L147-L152, and L196-L201), and the distinction between the method section and the result section is insufficient. Authors should carefully describe the manuscript for easy understanding.

Finally, there are some of the typos in this manuscript (i.e., "year/s" in P7L217 and "[24)" in P7L259). Authors should carefully check the manuscript before submission.

Reviewer 2 Report

Presented paper by Muleba L. is interesting and it deserves to be published however after some minor changes. 

L.33 - authors mentioned pathogens resistant to commonly used antibiotics - multiresistance phenomena is very interesting and authors could describe it a little bit more in the context of disinfection and hygiene

Fig.1. Where is statistical analysis in this figure? Standard deviation? how many repetitions were done? where are statistically significant differences between samples?

Table 1. statistical analysis?

Authors should also add some information in the Discussion part of the manuscript concerning selected bacteria strains survival on the skin surface, which influence effectiveness of sanitizers

Round 2

Reviewer 1 Report

Authors revised the manuscript point by point, but authors do NOT show (the mean values of) age among participants as mentioned in the first review. Because age as one of the basic characteristics may be associated with attitude regarding use of hand sanitisers. Authors should add the information regarding age of participants.

Author Response

Dear Reviewer,

Thank you for the clarification on the previous comment regarding the age of the participants.  "Because age as one of the basic characteristics may be associated with attitude regarding use of hand sanitisers. Authors should add the information regarding age of participants

Response:

We have added the age range, mean and standard deviation as well as sex in lines 258-262. And, have updated the discussion in lines 389-393.

Thank you very much for the comment.  We see the value of including the statistic.

Kind regards